# An Overview of Specific Considerations in Chronic Venous Disease and Iliofemoral Venous Stenting

**DOI:** 10.3390/jpm13020331

**Published:** 2023-02-15

**Authors:** Taimur Saleem

**Affiliations:** The RANE Center for Venous and Lymphatic Diseases, Suite 401, 971 Lakeland Drive, Jackson, MS 39216, USA; taimur@gmail.com

**Keywords:** intravascular ultrasound, iliofemoral vein stent, venous leg ulcers, quality of life, graduated compression stockings

## Abstract

Unlike arterial disease, chronic venous disease (CVD) is rarely life-threatening or limb-threatening. However, it can impose substantial morbidity on patients by influencing their lifestyle and quality of life (QoL). The aim of this nonsystematic narrative review is to provide an overview of the most recent information on the management of CVD and specifically, iliofemoral venous stenting in the context of personalized considerations for specific patient populations. The philosophy of treating CVD and phases of endovenous iliac stenting are also described in this review. Additionally, the use of intravascular ultrasound is described as the preferred operative diagnostic procedural tool for iliofemoral venous stent placement.

## 1. Introduction

Unlike arterial disease, chronic venous disease (CVD) is rarely life-threatening or limb-threatening. However, it can impose substantial morbidity on patients by influencing their lifestyle and quality of life (QoL). The symptoms of CVD can range from mild discomfort to QoL-impairing lower extremity pain. The clinical signs of CVD can range across a spectrum from edema, varicose veins, cutaneous hyperpigmentation, and lipodermatosclerosis to venous ulceration [1].

Traditionally, venous symptoms have been managed with conservative therapies. These include compression, ambulation, elevation, and wound care/antibiotics, in the case of venous ulceration. However, with the advent of minimally invasive therapies, these have been utilized increasingly in patients with CVD. For superficial venous disease, these minimally invasive therapies include ablation of the incompetent vein by different modalities. For chronic deep venous obstructive disease, these minimally invasive therapies include iliofemoral caval venous angioplasty and stenting. Open venous bypass surgery has a limited, secondary role today. Minimally invasive therapies are associated with quicker recovery, lesser post-operative pain, and lesser incidence of infections [2].

CVD can manifest in a variety of ways in individuals. CVD manifestations can be categorized according to the well-known Clinical-Etiology-Anatomy-Pathophysiology (CEAP) classification (see Table 1) [3]. Treatment should be tailored according to the individual patient, taking into account a variety of factors such as patient’s age, frailty, socioeconomic status, physiological condition, unique pathology, and anatomic peculiarities. The aim of this review was to focus on specific considerations for CVD and iliofemoral venous stenting.

## 2. Materials and Methods

Two databases (MEDLINE and Cochrane) were searched to obtain the most recent information on the topic of invasive (deep venous stenting for CVD) or conservative treatment of CVD (including compression stockings). The search was carried out in December, 2022. Search criteria included the word “iliac vein stenting” with a combination of the following words: CVD, compression, anticoagulation, May-Thurner syndrome, and iliac vein compression syndrome. Individual case reports or small case series, animal experiments, articles not written in English, and articles older than 20 years old were all excluded. Studies that focused on venous stenting in the setting of acute deep venous thrombosis (DVT) were also excluded. In addition, the paper should discuss specific patient populations (for example, pregnant females or octogenarians etc.) with a focus on specific considerations for CVD or iliofemoral venous stenting. References of these studies were reviewed for potential additional studies which were then searched for manually. Conference abstracts for which full length papers had not been published yet were also excluded. The search strategy is shown in Figure 1.

## 3. Results

### 3.1. Phases of Endovenous Stenting

Three important phases are identified with respect to venous stenting. First, the preoperative planning phase which entails outlining surgical objectives, understanding how they differ from goals of arterial surgery, and adequate knowledge of venous anatomy, landmarks, and anatomic variants. The second phase is intraoperative execution—this involves recognition of critical structures, performance of the procedure using best clinical judgment, and then adequately assessing the technical procedural success [4]. Third phase is the post-operative phase which includes institution of anticoagulation, when appropriate, and stent surveillance to detect stent malfunction or stent occlusion.

### 3.2. Philosophy of Treating CVD

Venous disease is not life threatening—therefore, initial management is inherently conservative and non-surgical. It is important to understand that there is no role for venous intervention in asymptomatic individuals [5]. In arterial disease, critical stenosis, even when asymptomatic, can portend an impending threat to regional perfusion in certain arterial beds. One well known example of this arterial phenomenon is carotid artery stenosis. In contrast, in venous disease, no such critical stenosis exists that threatens regional perfusion. A trial of conservative therapy for 3–6 months is appropriate for most patients with symptomatic venous disease. A step-by-step approach for the management of symptomatic and asymptomatic CVD is shown in Figure 2.

### 3.3. Elderly Population

Like atherosclerotic arterial disease, the incidence of CVD also increases with age. The more advanced manifestations of CVD (CEAP class C4–6) bear a unique burden on the geriatric population (patients older than 65 years of age). In the geriatric population, sepsis from lower extremity cellulitis occurs more frequently when compared to younger population groups [6]. Bacterial cellulitis is a common complication of CVD in the setting of dermatitis or venous ulceration [7]. Self-application of compression is cumbersome in the geriatric population group due to limited mobility, sedentary lifestyle, and comorbid conditions such as advanced arthritis and overall frailty [6]. In addition, skin care can be difficult, particularly when venous ulceration occurs. Frequent transportation to and from wound care centers is challenging for this population group as well when resources are already very limited. Frequent hospitalizations for cellulitis and venous ulcerations can ultimately lead to institutionalization of the elderly population [6]. Iliac venous stenting is a safe and effective prospect for such patients (including octogenarians and nonagenarians) and can provide an effective alternative to compression [6,8]. It can also be considered when compression fails. Several studies have now shown that iliac venous stenting not only expedites venous ulcer healing and cellulitis resolution but is also associated with high patency rates [7,9,10,11,12]. The procedure is minimally invasive and well tolerated by most of the geriatric population [7,8]. Technical considerations for stenting in the elderly remain the same as other patients. However, post-intervention, use of anticoagulation, or antiplatelet therapy should be done with caution due to risk of falls and bleeding. Risk-benefit ratio of such pharmacotherapy should be carefully weighed by the venous specialist in close consultation with the patient (shared decision making).

### 3.4. Socioeconomic Factors

Advanced manifestations of CVD can be a source of significant loss of productivity and chronic morbidity in younger patients [7]. A young patient with CVD represents a financial burden for the healthcare system as well. For example, consider a 35-year-old patient who earns a living by working on an oil rig off the coast of Mississippi, USA. He develops venous ulceration secondary to severe post-thrombotic syndrome (PTS) despite compliance with anticoagulation and compression stockings. His livelihood depends on him being able to carry out all the job functions regularly at the oil rig. Multiple absences to go to a wound care center on-shore due to his non-healing ulceration are a significant hindrance in this regard, both economically and socially. Iliac venous stenting offers an important therapeutic option for this patient in whom venous ulceration has occurred despite optimal conservative management. Healing of the venous ulceration after iliac venous stenting allows him to return to the oil rig where he is able to work uninterrupted. Additionally, barriers to his social life are improved.

### 3.5. Other Indications for Treatment

Main interventional indications for treatment of CVD include symptoms causing functional impairment such as disabling venous claudication, debilitating edema, moderate to severe venous pain, hemorrhage from varicose veins, non-healing ulcerations, recurrent infections or other sequelae of PTS and other complications of chronic venous hypertension that are refractory to conservative therapy. This generally includes patients in CEAP class C4 and above [11]. Interventional treatment of CEAP class C3 remains controversial amongst most leading venous authorities as edema resolution after venous stenting maybe partial in most cases. CEAP classification has been described in detail in Table 1.

### 3.6. Graduated Compression Stockings

Conservative therapy includes graduated compression stockings (GCS), leg elevation, ambulation, manual decongestive therapy, wound care, and antibiotics. The cornerstone of conservative therapy for CVD is GCS. The older treatment paradigm allotted compression therapy a primary therapeutic role. However, with the advent of iliac vein stenting, the therapeutic paradigm for CVD has expanded [13].

Several smaller trials have reported the symptomatic improvement in patients who wore GCS [14]. However, it did not reduce the incidence of post-thrombotic syndrome in some trials [15]. Moreover, in a Cochrane review from 2021 on the subject including 13 studies with 1021 participants, there was insufficient evidence to determine whether or not compression stockings are effective as the sole and initial treatment of varicose veins [16]. In another Cochrane review, compression appeared to increase healing rates of venous ulcers compared with no compression, and multi-component compression systems were found to be more effective than single-component systems. However, there was insufficient data to make secondary end point conclusions such as ulcer recurrence [17].

GCS is non-invasive in comparison to iliac venous stenting, which is minimally invasive. However, GCS attempts to control the end-effects of the venous pathology whereas iliac venous stenting corrects the source of the pathology (peripheral venous hypertension). The efficacy of compression varies widely because of the variety of compression products and bandaging techniques that are available. The caveat is that up to 50% of ulcers may recur despite the use of long term compression [13]. This recurrence rate is higher when compared to iliac venous stenting [18].

Another major limitation with compression is patient compliance. Non-compliance has been reported to exceed 50% in several reports [13]. Some of the reasons cited by patients for the discontinuation of GCS include its high cost, itching, cosmesis, edema exacerbation, allergic reactions to stocking material, and difficulty with application [19]. Some patients (up to 30%) were unable to specify a reason altogether [20].

One major question that arises ethically is whether patients who are non-compliant with GCS should be denied further treatment options such as iliac vein stenting. The situation is akin to an aortic aneurysm that meets criteria for repair in an individual who smokes. Further options should be offered to these patients because not all patients can tolerate GCS and because reasons for the non-compliance remain so poorly understood in current literature.

### 3.7. Thrombophilia Panel

Thrombophilia conditions are associated with an increased risk of venous and/or arterial thrombosis. Some of these conditions include antithrombin deficiency, protein C or protein S deficiency, presence of lupus anticoagulant (LA), factor VIII elevation, prothrombin and factor 5 gene mutations, and hyperhomocysteinemia. Other conditions that can provoke venous thromboembolism (VTE) include pregnancy, major surgery, immobilization due to severe illness or orthopedic injury, trauma, or hormone replacement therapy (HRT) [12]. At our center, the most common thrombophilia conditions encountered include factor VIII elevation (70%), hyperhomocyteinemia (25%), factor IX elevation (14%), and factor XI elevation (6%) [12].

The thrombophilia panel includes testing for the following factors at our center: homocysteine, prothrombin time (PT), international normalized ratio (INR), partial thromboplastin time (PTT), dilute russels viper venom time (DRVVT), thrombin time (TT), protein C, protein S, antithrombin III, prothrombin G20210A mutation, platelet count, factor V gene mutation, factor VIII, factor IX, factor XI, anticardiolipin (ACL) antibody, beta-2 glycoprotein antibodies, and LA [12].

There is little consensus in the literature about patient selection for thrombophilia testing. This testing may be considered in patients who are deemed high risk for recurrent VTE and in whom the initial VTE episode was unprovoked. Patients should be involved at all stages of the “shared decision making process”. Pre-test counselling should be provided to patients before thrombophilia panel testing is performed [12]. The following additional conditions may be considered where thrombophilia testing may be performed:When a risk of recurrent thrombosis can be identified via thrombophilia testing.When the management of asymptomatic family members who are carriers of the condition is impacted.Patient preference because he/she would like to better understand the etiology of the thrombotic event.When the risk of VTE recurrence is intermediate and obtaining the thrombophilia panel will help in decision making about long term anticoagulation [21].

Decision of choice of anticoagulation post-intervention is governed by a multitude of factors including surgeon preference, patient preference, history of recurrent deep venous thrombosis (DVT), agent affordability, prior patient experience including side-effects, co-existence of other medical conditions requiring anticoagulation (for instance, atrial fibrillation, valvular heart disease etc.), and presence of antiphospholipid syndrome [12].

### 3.8. Stenting across Inguinal Ligament

One of the fundamental principles of venous stenting in post-thrombotic patients is to stent from healthy vein to healthy vein in the presence of an adequate inflow and outflow for the stented conduit [5,22]. In patients with extensive post-thrombotic syndrome, this may occasionally require stenting across the inguinal ligament in order to achieve an adequate and healthy inflow for the stent. The decision to stent across the inguinal ligament should be individualized for every patient. This has been a point of contention amongst certain authorities; the fear being that extension of stents into the common femoral vein across the inguinal ligament will lead to increased rates of stent thrombosis, stent fractures, stent compression, or severe instent restenosis (ISR). Most of these assumptions were extrapolated from arterial literature. However, the landmark paper on this subject by Neglen et al. in 2008 [23] showed that braided stainless steel stents (n = 177) can be safely placed in the venous system across a moving joint such as the inguinal ligament without any significant impact on adverse outcomes. This study also noted that the patency rate of the venous stent was more so dependent upon the etiology and nature of the venous obstruction [23]. More recently, this variable has been examined for the dedicated venous nitinol stents. The rates of stent fractures following nitinol stents were negligible or zero for the various stents, even when crossing the inguinal ligament in various clinical studies: ABRE stent [24], VICI stent [25], Venovo stent [26], and Zilver Vena [27].

### 3.9. Intravascular Ultrasound

Intravascular ultrasound (IVUS) has made the art and science of treating CVD more precise, accurate, and personalized. It provides an allowance for the individualization of each patient’s treatment plan. Being the gold standard, IVUS provides high quality, real-time cross-sectional venous anatomy and is becoming increasingly available for venous interventions. Measurements obtained via IVUS provide guidance in iliofemoral venous stenting by providing ideal proximal and distal landing zones for the stents. In addition, IVUS provides crucial information for accurate stent sizing. Additionally, IVUS can be used in patients with advanced renal failure or severe allergy, to contrast. Through the use of IVUS, the following mishaps can be avoided: ‘jailing’ of the contralateral iliac vein orifice and ‘jailing’ of the ipsilateral profunda vein orifice [28,29].

Venography, even with multi-planar technique, can underestimate the presence and severity of iliac venous stenosis [30]. In one study, the median maximal area stenosis was noted to be significantly higher with IVUS than venography (69% vs. 52%, *p* < 0.0001). Iliac-caval confluence, a crucial technical landmark in venous stenting, correlated between venography and IVUS in only 15% of the patients. Therefore, sole reliance on venography can lead to undertreatment of venous lesions. Similarly, venography and IVUS correlated with each other in only 26% of cases as far as the distal landing zone for the stent was concerned [31].

### 3.10. In-Stent Restenosis

In-stent restenosis (ISR) causes stent malfunction that can be unavoidable in certain cases (Figure 3). There are two important considerations with respect to ISR in patients with venous stents. Firstly, about 20–40% ISR is common in most venous stents. Secondly, ISR very rarely leads to complete stent occlusion (<10%). This in turn has led to two modifications in stenting techniques: firstly, slightly oversizing of Wallstents is recommended so that it can somewhat compensate for future development of ISR while also allowing for more aggressive balloon dilatation. Secondly, there is no role or recommendation for the prophylactic balloon dilatation of stents with ISR in asymptomatic individuals. ISR is affected by two main factors: a stent inflow area <125 mm^2^ and shear rate >100 s^−1^. Tapered stent profile may help with the latter. Drug eluting balloons and stents also represent a future area of research in the prevention of ISR in the iliofemoral venous system but the large surface area involved compared to arterial or coronary systems must be carefully considered [32,33].

### 3.11. Adequate Sizing of Venous Stents

In the vasculature (both venous and arterial systems), adequate stent sizing is of utmost importance. In the arterial system, stents are usually slightly undersized for fear of dissection or perforation. In the venous system, undersizing stents will likely lead to iatrogenic stenosis and stent failure. The caliber (absolute cross-sectional area) of the iliac venous outflow controls peripheral venous pressure [34]. Therefore, venous stents should mirror normal venous anatomy to adequately decompress peripheral venous hypertension. The optimum sizing for iliac venous stents based on data derived from flow equations, IVUS planimetry, and Poiseuille equation in non-diseased venous segments in healthy volunteers has been described in detail previously. These stent diameters are: common iliac vein: 16–18 mm (area: 200 mm^2^), external iliac vein: 14 mm (area: 150 mm^2^), and common femoral vein: 12 mm (area: 125 mm^2^) [34].

Undersizing venous stents will cause residual symptoms despite stent patency demonstrated on imaging studies such as venography. For example, a 14-mm stent placed in the CIV represents an iatrogenic area stenosis of 25% from the time the stent is placed and this will lead to residual symptoms. ISR, a commonly prevalent problem in venous stents, will cause further area reduction and lead to recurrence of symptoms in the future. This iatrogenic stenosis due to stent undersizing will lead to stent failure and can be difficult to correct, necessitating multiple reoperations [5,34,35]. Additionally, undersized stents can occlude or migrate. With the Wallstents, slight oversizing (1–2 mm) is generally recommended. With the newer dedicated nitinol stents, such oversizing is not necessary. At the time of initial stent implantation, ballooning should be restricted to the optimal rated caliber of the stent itself.

### 3.12. Anticoagulation and Antiplatelet Regimens

Antiplatelet and anticoagulation regimens after venous stenting remain controversial. There are no large trials available for guidance in this particular arena. Most of the evidence is heterogeneous, anecdotal, or based on variable consensus guidelines. In a study on 62 patients by Endo et al., [36], stent patency was best predicted by concomitant antiplatelet and anticoagulation therapy rather than anticoagulation alone. Some authorities believe that not all anticoagulation is the same. However, at least one retrospective study with 100 consecutive patients has shown adequate primary patency rates when using direct oral anticoagulation therapy (DOAC), compared with those treated with warfarin or low molecular weight heparin [37]. According to an International Delphi Consensus, anticoagulation was the preferred treatment during the first 6–12 months following venous stenting for a compressive iliac vein lesion. There was no agreement on the long term role of antiplatelet therapy in venous stenting [38]. In a systematic review by Veyg et al. in 2022, there was no apparent correlation between medication used and stent patency or subjective patient outcomes. This review called in to question whether anticoagulation was necessary at all following stenting of NIVL type lesions because of similar excellent outcomes among the different agents that were used [39]. However, there was no control group where no pharmacotherapy was utilized. Lastly, the effect of triple therapy has been rarely studied. In one study on 87 patients, there was a reduction of in-stent restenosis/thrombosis events when triple therapy was used compared to antiplatelet only regimens [40]. However, there is also a higher risk of bleeding events associated with triple therapy when compared to antiplatelet only regimens and therefore, this should be utilized with caution, especially in the geriatric population.

### 3.13. Diabetes and Venous Stenting

Diabetes is not a traditional risk factor for venous disease; in contrast to arterial disease. However, some population-based studies have reported increased risk of DVT and pulmonary embolism in diabetics compared to the general population [41,42]. Although diabetes is a frequent comorbidity in patients with CVD, the exact correlation between the two is not yet known [43]. Similarly, unlike in the arterial system, where the presence of diabetes may accelerate the incidence of in-stent restenosis (ISR) [44,45], the effect of diabetes on ISR in venous stents is not clearly known.

### 3.14. May-Thurner Syndrome and Compressive Iliac Vein Lesions

Compression of the left common iliac vein by the overlying right common iliac artery is commonly found in many asymptomatic individuals in the general population. However, only a subset of patients develops symptoms of CVD despite having “May-Thurner anatomy”. In one study on 50 emergency room patients presenting with abdominal pain, computed tomography scans were reviewed. These patients did not have any venous disease symptoms attributable to iliac vein compression. However, surprisingly, almost a quarter of patients had greater than 50% compression and 66% patients had greater than 25% compression of the left common iliac vein without symptoms [46]. Many individuals in the general population may have silent iliac vein stenosis on cross-sectional imaging—however, the pathology of iliac vein stenosis is permissive. This means that a second insult is needed to destabilize the “venous homeostasis” and cause the symptoms of venous disease to manifest. Such insults include infection, thrombosis, trauma, edematogenic medications, and development of reflux. [47].

Chronic compression can lead to inflammation within the lumen of the vein, resulting in intimal fibrosis [48]. This ultimately leads to a localized flow disturbance and peripheral venous hypertension (Figure 4). The mean age of presentation is 42.6 ± 16.9 years [49]. However, contrary to the traditional description, iliac vein compression syndrome can occur in both genders, on both sides, and at any age [47]. Several other anatomical variants have been reported in literature and all of these lesions are known as NIVLs [50].

### 3.15. Various Types of Venous Stents

Initially, Wallstents were used in the venous system; although they were not approved for this particular indication. More recently, dedicated nitinol venous stents are available for use in the iliofemoral venous system [34,51,52]. Wallstent is made of Elgiloy material. Nitinol, on the other hand, is an alloy of nickel and titanium. Details regarding various types of venous stents and their patencies are shown in Table 2 and Table 3. Of the four dedicated nitinol stents available, two (VENOVO and VICI) were voluntarily recalled due to issues related with deployment system and stent embolization respectively. The VENOVO stent is now available again in the market for the treatment of CIVO.

With the Wallstents, a reintervention rate of 15% is not uncommon and an occlusion rate of about 3% has been reported in large series [5,32]. Reinterventions for venous stents may include angioplasty for stent compression or ISR, extension of the stent cephalad or caudad, recanalization or thrombectomy/thrombolysis of an occluded stent. Additional data are being collected on the newer dedicated nitinol venous stents on these parameters.

### 3.16. Iliac Vein Stenting without Venography (Dyeless Iliac Vein Stenting)

At most venous centers across the country, venography is utilized with or without intravascular ultrasound to guide endovenous stenting. We reported our experience in a subset of 31 limbs in whom venography was not utilized and endovenous stenting was carried out only with the help of IVUS guidance. This study showed that IVUS had a high clinical yield in patients in whom signs and symptoms of CVD were severe enough to merit further diagnosis and intervention [28].

### 3.17. Extension of the Iliac Vein Stent into the Profunda Vein

The extension of iliac vein stents into the common femoral vein is often required, especially in in post thrombotic limbs, to fully correct venous pathology. However, rarely, an extension of the iliac vein stenting is needed into the profunda vein as well. We showed in a series of 20 limbs that this procedure is rarely required but useful for stent salvage and symptom relief [53].

### 3.18. Effect of Iliac Vein Stenting on Reflux

We have previously shown that iliac vein stenting either leads to improvement or stabilization of reflux in the stented limbs. This may be due to removal of pressure on the valve station after stenting and subsequent decrease in the size/less distention of the vein. In addition, most limbs tolerated untreated reflux clinically well across the majority of clinical parameters. Therefore, as a stand-alone procedure, venous stenting produces adequate clinical outcomes [10].

## 4. Discussion

### 4.1. Types of Lesions

Venous stenting for chronic iliofemoral venous obstruction (CIVO) has now replaced open surgery as the standard of care in symptomatic patients who are not responsive to a trial of conservative therapy for at least 3–6 months. The interventionist can encounter either PTS lesions or non-thrombotic iliac vein lesions (NIVL) in patients with CVD. Both lesions have differing characteristics and etiologies. Similarly, results of venous stenting in both groups of patients differ despite the use of similar techniques and paraphernalia/stents [54]. IVUS has allowed us to better understand the behavior and characteristics of these lesions. Post-stenting, anticoagulation is usually not deemed necessary in NIVL patients but is frequently employed in patients with PTS lesions [55]. Initially the Wallstent was the only stent used in the venous system; its use was off-label. However, the venous landscape has now matured significantly and several other dedicated nitinol venous stents are now available in the market for the purpose of stenting for CIVO. Also, the Wallstent is now FDA (US Food and Drug Administration) approved for use in the human venous system [56].

### 4.2. Patient Selection and Criteria for Stenting

Patient selection is paramount in the treatment of patients with CIVO [57]. As mentioned earlier, such patients should have persistence of life-style limiting symptoms despite a trial of conservative therapy. They should have signs and symptoms consistent with CVD in addition to radiological imaging supporting a diagnosis of venous stenosis. This diagnosis should be confirmed intraoperatively with the use of IVUS. Sole reliance on venography for the confirmation of venous stenosis should be discouraged as it will under-treat many patients and miss detection of many lesions. In addition, 50% stenosis criteria based on comparison to normal venous segments (contralateral or ipsilateral) should not be used. This is a concept which has been extrapolated from arterial literature and should not be applied to venous measurements. This is because iliac veins are unique in that the pathology can include Rokitansky lesions—these are diffuse, long lesions without any focal clues and will be missed on venography. IVUS provides a definitive diagnosis of Rokitansky lesions. Comparison of measured areas should be made against normal minimal luminal areas which have been described earlier (CIV 200 mm^2^, EIV 150 mm^2^, CFV 125 mm^2^) and which were calculated based on flow equations and IVUS planimetry data [58]. Additionally, there is no critical threshold stenosis in veins, unlike arteries. Veins do not exhibit powerful compensatory vasodilation in response to stenosis. Peripheral venous hypertension arising from CIVO is non-linear [57]. In addition to long segment stenosis, there can be skip lesions and bilateral disease—therefore, the use of comparative 50% technique is discouraged (Figure 5) [57].

### 4.3. Other Special Groups of Patients/Miscellaneous Considerations

#### 4.3.1. Central Neuromuscular Disorders

Patients with central neuromuscular disorders can have significant leg swelling and pain. These patients include those with conditions such as multiple sclerosis and Parkinson’s disease. We investigated venous stenting in these patients and found a significant improvement in multiple clinical parameters in these patients including edema grade, pain score, venous clinical severity score (VCSS), and ulcer healing. However, the rate of reinterventions in these patients after initial venous stenting was high (53%); this was likely due to primary calf pump dysfunction [9].

#### 4.3.2. Klippel-Trenaunay Syndrome

Klippel-Trenaunay syndrome (KTS) is a congenital hemangiolymphatic mesenchymal malformation syndrome that includes varicose veins, capillary and venous malformations, lymphatic abnormalities, and hypertrophy of various connective tissue elements [11]. We have described the outcomes of venous stenting in this rare syndrome and noted significant improvement in clinical parameters such as quality of life, pain score, venous clinical severity score, grade of edema, and ulcer healing [11]. Treatment should be individualized to each patient’s presentation and symptoms and should be carried out in a step-wise manner.

#### 4.3.3. Obese Patients

Self-application of compression stockings is difficult in the obese patient as many obese patients are unable to reach their feet [59]. Obese patients have been noted to have more severe peripheral venous hypertension than their non-obese counterparts [60]. In one study, bilateral clinical manifestations of CVD were twice as common in the obese patient subset compared with the non-obese patient group (28% vs. 14%, *p* = 0.0007). NIVL or PTS lesions, as seen in non-obese patients, were seen in 89% of cases on IVUS while in the remainder of patients (11%), compression of venous outflow related to high intra-abdominal pressure was contributory to CVD. Improvements in clinical parameters after stenting included improvements in pain, swelling, resolution of dermatitis, quality of life and ulcer healing [60,61].

#### 4.3.4. Pregnancy

There are several case series where authors have documented the experience of patients who became pregnant after placement of iliofemoral caval stents for NIVL or PTS lesions [62,63]. No events of DVT recurrence, symptomatic pulmonary embolism, or stent occlusion occurred as a consequence of pregnancy in these patients [62]. Therefore, iliac-caval stenting is not contraindicated in women of reproductive age. There is no damage noted to the iliofemoral caval stents from the gravid uterus [62]. Low-molecular-weight heparin (LMWH) should be considered, with dosing individualized to each patient, for stent thrombosis prophylaxis during pregnancy in patients with a history of iliofemoral caval stenting, as pregnancy itself is a pro-thrombotic condition [63].

#### 4.3.5. Post-Menopausal Patients

Leg swelling in post-menopausal women is believed by many practitioners to be amorphous and polycentric in origin. Many of these patients are placed on empiric diuretics without substantial improvement in symptoms [64]. However, in our own experience, we have observed obstructive venous pathology in these patients using IVUS. Significant improvement in swelling, pain, and quality of life was noted after iliac venous stenting in this patient population [64].

#### 4.3.6. Patients with Femoral Vein Occlusion

In cases of femoral vein occlusion, as occurs in thrombosis, rapid axialization of the profunda femoris vein occurs to compensate for collateral flow. However, iliac vein collateralization is less efficient and scarcer in contrast to femoral vein collateralization [65,66]. In patients in whom the femoral vein is occluded, iliac vein stenting produces significant clinical improvement in symptoms such as swelling, pain, and ulcer healing. We also found that the great saphenous vein has little contribution to the overall pathology in this process and can be safely ablated in the appropriate clinical scenario in patients with femoral vein occlusion and iliac outflow venous obstruction [65]. Additionally, we have demonstrated that stent extension into the profunda vein from the iliac vein can be carried out safely in selective cases, although it is infrequently required [53].

#### 4.3.7. Bilateral Iliac Vein Stenting

Bilateral iliac vein stenting at the iliac-caval confluence can be carried out using a variety of techniques: double barrel stenting (placement of two stents side by side), apposition technique (apposition of a stent as close as possible to a stent previously placed across the iliac-caval confluence), inverted Y stenting technique (inverted Y stenting through a fenestration created through the side of a previously placed contralateral stent), and interdigitation of bilateral Zenith-stents (Cook Medical, Bloomington, IN, USA) technique [67]. However, our recent experience has shown that in up to 95% patients with bilateral chronic iliofemoral venous obstruction, treatment of worse ipsilateral limb results in symptomatic improvement of the contralateral limb. This phenomenon is likely related to the off-loading of the pelvic and inguinal collaterals. Therefore, in selective patients, sequential rather than simultaneous bilateral iliac vein stenting approach should be pursued [68].

## 5. Limitations

The first limitation of this study is incorporation of two databases in the search criteria. Secondly, because of the nature of the research query, a systematic review or meta-analysis could not be performed. Instead, an overview has been presented based on the author’s extensive experience and expertise in the field of venous disease.

## 6. Conclusions

CVD can impose significant morbidity on patients by impairing their quality of life and daily activities. Treatment should be individualized in every patient. A trial of conservative management is appropriate in most patients with CVD. However, it is important to understand that some patients maybe intolerant of conservative therapeutic measures (for example, octogenarians and nonagenarians). Iliofemoral stenting appears to be a safe and effective modality that produces durable clinical improvements in patients with CVD. Experience in a wide variety of patient subsets has been presented including patients with different medical conditions and patients of different age groups, physiology, and pathology. IVUS is an important tool that tailors the treatment plan according to the unique patient profile and should be used in every deep venous intervention.

## Figures and Tables

**Figure 1 jpm-13-00331-f001:**
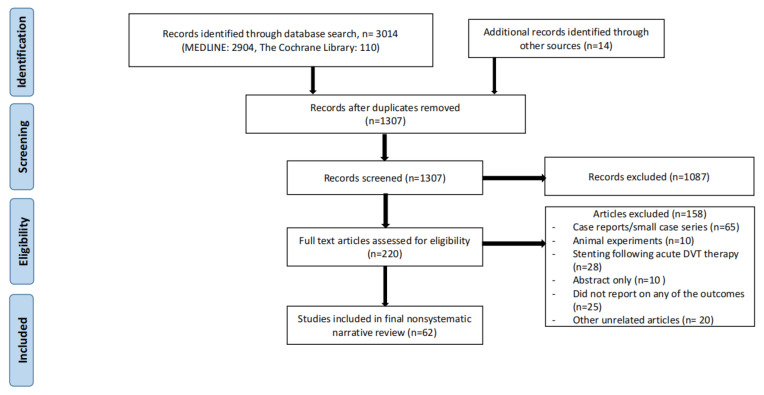
Search strategy for nonsystematic narrative review.

**Figure 2 jpm-13-00331-f002:**
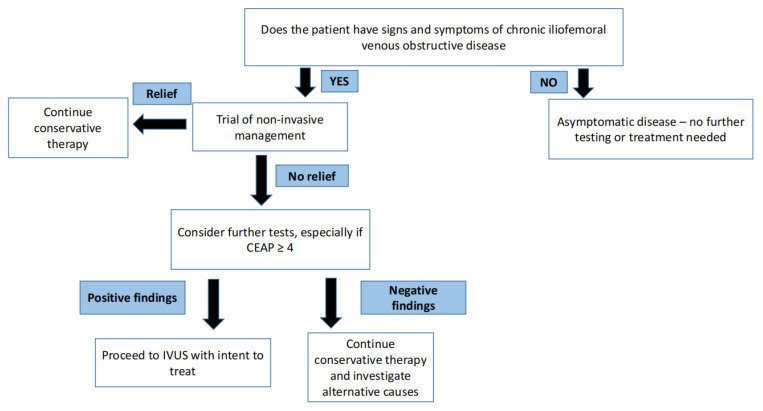
Philosophy of treating symptomatic and asymptomatic chronic venous disease.

**Figure 3 jpm-13-00331-f003:**
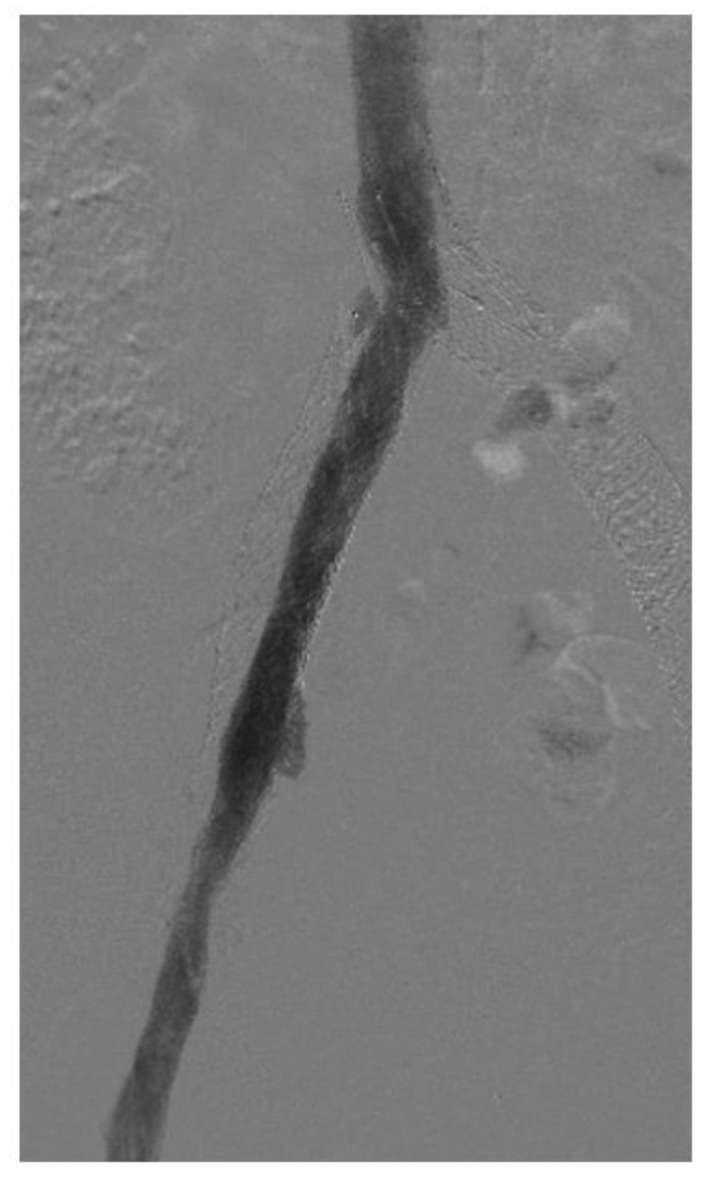
Venogram showing in-stent restenosis in the stent (about 50%). Notice that the contrast does not completely fill the stent lumen.

**Figure 4 jpm-13-00331-f004:**
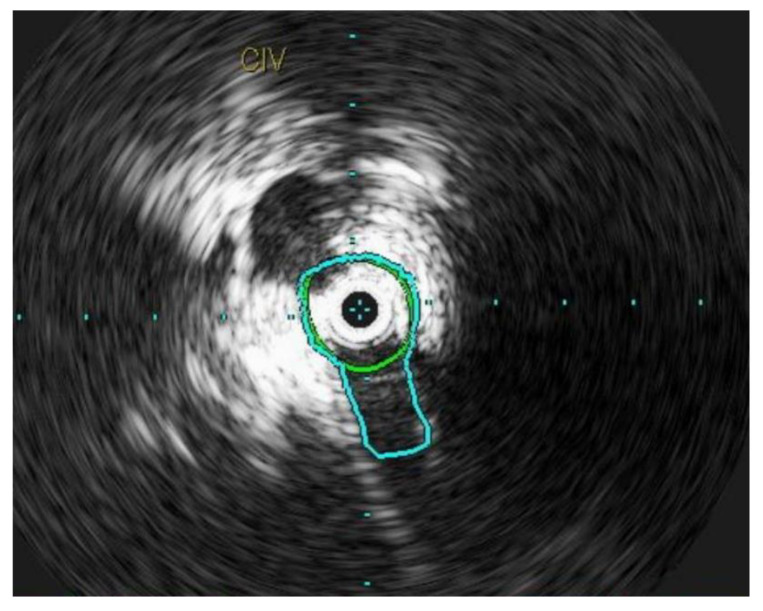
L common iliac vein is compressed between the vertebral body and the artery and appears as a narrow slit like structure.

**Figure 5 jpm-13-00331-f005:**
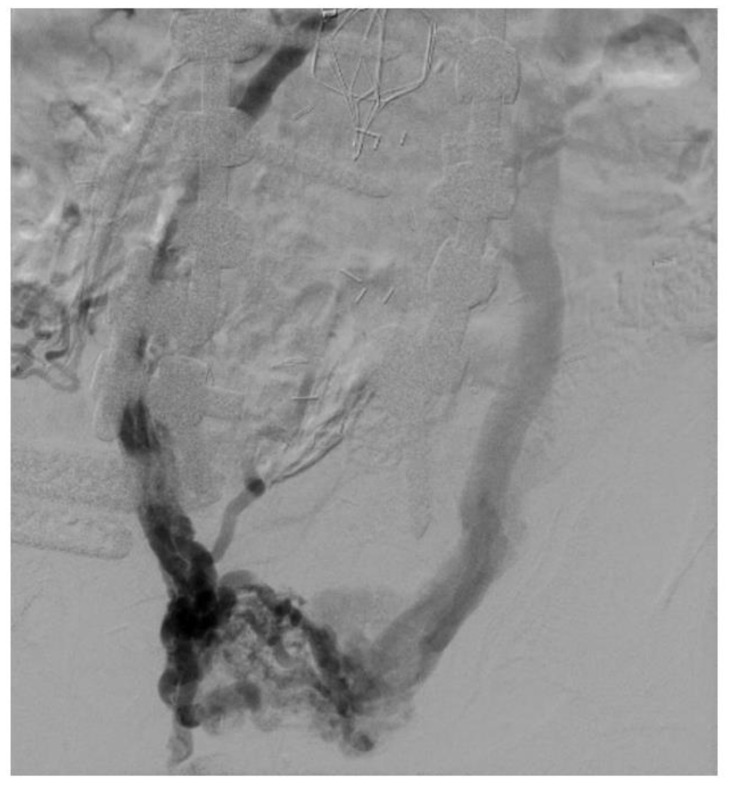
Large pelvic collaterals seen in a patient on venography with bilateral iliac vein occlusion. Also seen is an IVC filter with IVC occlusion.

**Table 1 jpm-13-00331-t001:** CEAP Classification System.

C (Clinical), E (Etiology), A (Anatomic), P (Pathophysiological)	
C0	No visible or palpable signs of venous disease
C1	Telangiectasias or reticular veins
C2	Varicose veins
C2r	Recurrent varicose veins
C3	Edema
C4	Changes in skin and subcutaneous tissue secondary to chronic venous disease
C4a	Pigmentation or eczema
C4b	Lipodermatosclerosis or atrophie blanche
C4c	Corona phlebectatica
C5	Healed ulcer
C6	Active venous ulcer
C6r	Recurrent active venous ulcer

**Table 2 jpm-13-00331-t002:** Details of various types of venous stents.

Type of Stent	Company	FDA Approval	Stent Design	Delivery System
Wallstent	Boston Scientific Corporation	2020	Braided	Coaxial
Abre	Medtronic	2020	Open cell	Triaxial
Venovo	BD Interventional	2019	Open cell	Triaxial
VICI	Boston Scientific Corporation	2019	Closed cell	Coaxial
Zilver Vena	Cook Medical	2020	Open cell	Coaxial

**Table 3 jpm-13-00331-t003:** Patencies of various types of venous stents.

Type of Stent	Trial	Patients (N)	Patency
Wallstent	No formal clinical trial; plethora of data available from large retrospective case series	-	Primary (5-year): 67% Primary-assisted (5-year): 89% Secondary (5-year): 93%
Abre	ABRE	200	Primary (2-years): 86.2%
Venovo	VERNACULAR	170	Primary (3-years): 79.5%
VICI	VIRTUS	200	Primary (3-years): 71.7%
Zilver Vena	VIVO	243	Primary (3-years): 90.3%

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
