# Peer review of "An Overview of Specific Considerations in Chronic Venous Disease and Iliofemoral Venous Stenting"

_jpm, 2023, doi:10.3390/jpm13020331_

Round 1

Reviewer 1 Report

I reviewed an article entitled “An overview of personal considerations in chronic venous disease and iliofemoral venous stenting”. The overall manuscript was well written. I have some comments.

Major comments

1.       This was a systematic review. However, the methods were poorly described. Author did not include or at least mention pivotal clinical trials in relevant area, for example, GCS trials.

2.       Author should provide the complete inclusion criteria for the studies, date and time of the search, key words in each database and number of retrieved articles.

3.       Author should describe the scope and objective of the review.

4.       I am not sure about the difference between Results part and Discussion part. Should elderly people be moved to subgroup population?

5.       Thrombophilia testing seems not to be relevant to the topic. I would suggest to remove this part.

6.       The anticoagulant regimen was not thoroughly described. Author should describe more extensive, since, this remains controversial.

7.       Any studies regarding May-Thurner syndrome

8.       Authors use PTS, CVD alternatively. Should the term be consistent.  

Author Response

Reviewer 1

Reviewer comments:

Major comments

  1. This was a systematic review. However, the methods were poorly described. Author did not include or at least mention pivotal clinical trials in relevant area, for example, GCS trials.

Response: This is not a systematic review. It is a nonsystematic narrative review. Therefore, the search strategy is different than for a typical systematic review. Information on GCS has been added to the relevant section.   

  1. Author should provide the complete inclusion criteria for the studies, date and time of the search, key words in each database and number of retrieved articles.

Response: This has been provided in figure 1.

  1. Author should describe the scope and objective of the review.

Response: This has been described already. Please refer to the methods section.

  1. I am not sure about the difference between Results part and Discussion part. Should elderly people be moved to subgroup population?

Response: Subgroup population has been mainly discussed in discussion part of the paper. Elderly population is believed to be at a suitable place in the manuscript currently and moving it will disrupt the flow of the report. Therefore, we elected not to move it.

  1. Thrombophilia testing seems not to be relevant to the topic. I would suggest to remove this part.

Response: If anticoagulation is to be discussed and included, then thrombophilia testing becomes relevant as well. Therefore, it has not been removed.

  1. The anticoagulant regimen was not thoroughly described. Author should describe more extensive, since, this remains controversial.

Response: Anticoagulation and antiplatelet regimen has been discussed in more detail as it remains controversial.

  1. Any studies regarding May-Thurner syndrome

Response: Studies on May-Thurner syndrome have been included.

  1. Authors use PTS, CVD alternatively. Should the term be consistent.  

Response: These terms are different and not used alternatively. That has been incorrectly inferred.

Reviewer 2 Report

This review by Dr. Saleem (single author) about the invasive treatment of the iliofemoral chronic venous disease represents an interesting topic. You can immediately understand he is familirzed with the subject and I believe the interventional and non-interventional community needs such a "to the point" article. Indeed the review fields more like a commentary than of a state of the art review but this can be easily changed/improved. The essence remains. The flow and the language of the article is good, more like a textbook review - as the author correctly classifies it as a narrative review with a "personal touch". 

Comments

1. "Personal touch" should remain professional, as in objective observations not mere opinions even if the author has a high expertise in that specific field. I am not saying that this is the case here but the author, for example, may wish to remove the "personal considerations" part from the title. In the end, this is a review. 

2. Last sentence from the Introduction: "In this review, we want to focus" could be changed for a better option, for example "The aim of this review was"

3. The sentence before that: "taking into account a variety of factors" - maybe continue it with telling what factors - "such as...". 

4. Section 2 - Add a flowchart of the selection of the studies should be added (how many found, how many included, how many excluded, why, how many left, etc). 

Results - Section 3.0 - add this section on what are the results from your data scooping and what are the main findings. Add a table here on all the types of stent used in venous angioplasty, add variables such a as types (venous stents, self expandable, balloon expandable), companies (Zilver, etc), date of FDA approval/introduction, material (cobalt chromium, etc). 

Add then a paragraph with the general long term patency with venous stents and the rate of target lesion failure and revasc.

5. Section 3.2. Phylosohpy - maybe draw a diagram or a table on the step-by-step approach for a symptomatic and non-symptomatic CVD. 

Add what antiplatelet treatment is recommended for vein stenting.

6. Section 3.3 What are the particularities (med therapy and stenting techniques) in the elderly?

7. Section 3.4 - add a phrase also on the fact that a young patient with CVD represents a financial burden for the healthcare system as well. 

8. Section 3.5 - i think it's better for who reads this to add and reference the CEAP classification. 

9. Discussion. As I will mention below, maybe a re-arragement of your sections would make this article more friendly. 

Are there any particularities in patients with diabetes that receive venous stenting? Higher risk of restenosis similar to perihpereal artery disease stenting? You can add 2-3 sentences on this matter. Please cite the following works: DOI: 10.1155/2022/4196195 and DOI: 10.3390/ijerph19169801

  •  
  •  

MAJOR issues:

1. I would give up the methods-results-discussion design as this is a narrative review. Try to find some better names for the sections. 

2. The paper severely lacks visuals (in fact, there are no images). Please add from your armamentarium some nice images with cases of venous stenting. You can even add a section on "personal experience" or "center's experience".  

Overall, a nice overview that deserves publication after some revision. I congratulate the author for putting this together. 

Author Response

Reviewer 2

This review by Dr. Saleem (single author) about the invasive treatment of the iliofemoral chronic venous disease represents an interesting topic. You can immediately understand he is familirzed with the subject and I believe the interventional and non-interventional community needs such a "to the point" article. Indeed the review fields more like a commentary than of a state of the art review but this can be easily changed/improved. The essence remains. The flow and the language of the article is good, more like a textbook review - as the author correctly classifies it as a narrative review with a "personal touch". 

Response: Thank you. As correctly pointed out, this review is a narrative nonsystematic expert review. It is not meant to be a state of the art or exhaustive systematic review on the subject.

Comments

  1. "Personal touch" should remain professional, as in objective observations not mere opinions even if the author has a high expertise in that specific field. I am not saying that this is the case here but the author, for example, may wish to remove the "personal considerations" part from the title. In the end, this is a review. 

Response: The word “Personal” has been removed from the title.

  1. Last sentence from the Introduction: "In this review, we want to focus" could be changed for a better option, for example "The aim of this review was"

Response: This has been changed from “In this review, we want to focus” to “The aim of this review was”.

  1. The sentence before that: "taking into account a variety of factors" - maybe continue it with telling what factors - "such as...". 

Response: “such as patient’s age, socioeconomic status, physiological condition, unique pathology and anatomic peculiarities.”

  1. Section 2 - Add a flowchart of the selection of the studies should be added (how many found, how many included, how many excluded, why, how many left, etc). 

Results - Section 3.0 - add this section on what are the results from your data scooping and what are the main findings. Add a table here on all the types of stent used in venous angioplasty, add variables such a as types (venous stents, self expandable, balloon expandable), companies (Zilver, etc), date of FDA approval/introduction, material (cobalt chromium, etc). 

Add then a paragraph with the general long term patency with venous stents and the rate of target lesion failure and revasc.

Response: Flowchart or the studies has been added (see figure 1).

Tables have been added as recommended.

Paragraph on patency has been added and this information is also present in the tables now.  

  1. Section 3.2. Phylosohpy - maybe draw a diagram or a table on the step-by-step approach for a symptomatic and non-symptomatic CVD

Add what antiplatelet treatment is recommended for vein stenting.

Response: A diagram has been added in the philosophy section.

Section on antiplatelet regimen has been added as recommended.

  1. Section 3.3 What are the particularities (med therapy and stenting techniques) in the elderly?

Response: Stenting techniques are not any different in the elderly. Anticoagulation and antiplatelet therapy should be used with caution.

  1. Section 3.4 - add a phrase also on the fact that a young patient with CVD represents a financial burden for the healthcare system as well. 

Response: This phrase has been added.

“A young patient with CVD represents a financial burden for the healthcare system as well.”

  1. Section 3.5 - i think it's better for who reads this to add and reference the CEAP classification. 

Response: CEAP classification has been described in detail in table 1.

  1. Discussion. As I will mention below, maybe a re-arragement of your sections would make this article more friendly. 

Are there any particularities in patients with diabetes that receive venous stenting? Higher risk of restenosis similar to perihpereal artery disease stenting? You can add 2-3 sentences on this matter. Please cite the following works: DOI: 10.1155/2022/4196195 and DOI: 10.3390/ijerph19169801

Response:

Diabetes is not a traditional risk factor for venous disease. Effect on ISR in venous stents is not clearly known currently. Section on diabetes and venous stenting has been added.

MAJOR issues:

  1. I would give up the methods-results-discussion design as this is a narrative review. Try to find some better names for the sections. 

Response: These sections are named according to the journal format. Renaming them will create confusion and unfamiliarity for the esteemed readers.

  1. The paper severely lacks visuals (in fact, there are no images). Please add from your armamentarium some nice images with cases of venous stenting. You can even add a section on "personal experience" or "center's experience".  

Response:  Figures and tables have been added – thank you.

Round 2

Reviewer 2 Report

I am pleased to see such a quick and substantial revision. The author addressed all my comments in a point-by-point manner. Unfortunately, I am not able to see the Figures as the are not attached to the manuscript I have received but I trust the author they are interesting and suitable. I have only 1 comment left - I still believe you should remove the "Methods-Results-Discussion" subtitles and add your subtitles according to your subject, in a more logical way. This is a narrative review and the journal will permit you that. 

Other than that, I have nothing to add.